# Coevolution of Snake Venom Toxic Activities and Diet: Evidence that Ecological Generalism Favours Toxicological Diversity

**DOI:** 10.3390/toxins11120711

**Published:** 2019-12-06

**Authors:** Emma-Louise Davies, Kevin Arbuckle

**Affiliations:** Department of Biosciences, College of Science, Swansea University, Swansea SA2 8PP, UK; eldavies94@outlook.com

**Keywords:** evolution, ecology, prey diversity, predator-prey coevolution, neurotoxicity, coagulotoxicity, cytotoxicity, nephrotoxicity

## Abstract

Snake venom evolution is typically considered to be predominantly driven by diet-related selection pressures. Most evidence for this is based on lethality to prey and non-prey species and on the identification of prey specific toxins. Since the broad toxicological activities (e.g., neurotoxicity, coagulotoxicity, etc.) sit at the interface between molecular toxinology and lethality, these classes of activity may act as a key mediator in coevolutionary interactions between snakes and their prey. Indeed, some recent work has suggested that variation in these functional activities may be related to diet as well, but previous studies have been limited in geographic and/or taxonomic scope. In this paper, we take a phylogenetic comparative approach to investigate relationships between diet and toxicological activity classes on a global scale across caenophidian snakes, using the clinically oriented database at toxinology.com. We generally find little support for specific prey types selecting for particular toxicological effects except that reptile-feeders are more likely to be neurotoxic. We find some support for endothermic prey (with higher metabolic rates) influencing toxic activities, but differently from previous suggestions in the literature. More broadly, we find strong support for a general effect of increased diversity of prey on the diversity of toxicological effects of snake venom. Hence, we provide evidence that selection pressures on the toxicological activities of snake venom has largely been driven by prey diversity rather than specific types of prey. These results complement and extend previous work to suggest that specific matching of venom characteristics to prey may occur at the molecular level and translate into venom lethality, but the functional link between those two is not constrained to a particular toxicological route.

## 1. Introduction

Venom has evolved many times throughout the animal kingdom, with most major lineages having evolved it at least once [1,2]. The frequent convergent evolution of venom systems is probably a consequence of two attributes. Firstly, all venoms are intricately tied to antagonistic coevolution [2] and this scenario is widely considered to favour the origin and elaboration of trait diversity [3,4,5]. Secondly, venom can fulfil multiple functions which are core to fitness, such as antipredator defence and prey capture, either because it is incidentally suited to functions other than those it has primarily been selected for [2], or because the venom system has specifically been selected for multiple functions [6,7]. The end result is that animal venoms are both taxonomically and functionally diverse in addition to the substantial chemical diversity widely recognized amongst toxinologists [1].

Amongst animal venoms, those of caenophidian (‘advanced’) snakes are the best studied, largely as a result of the global medical importance of venomous snakebite [8,9]. Despite the defensive nature of snakebites inflicted on humans, this is a secondary function in the sense that the main selective driver of snake venom evolution is prey subjugation [1,2,10]. Notwithstanding a range of other adaptive and non-adaptive processes that may have contributed to variation in snake venoms [11,12], a key role of diet-related selection is supported by a variety of types of evidence.

The availability of different prey species will vary geographically, and their availability to an individual snake will vary with ontogeny, either via ontogenetic shifts in diet or simply in predation-relevant prey characteristics such as size. Consequently, several studies have looked for geographic or ontogenetic variation in venom composition or toxicity that may be explained by variation in prey. For instance, Daltry et al. [10] found that variation in the venom of the Malayan pitviper (*Calloselasma rhodostoma*) was related to both geographic and diet variation but that diet was the controlling factor, with geographic variation in venom as a result of geographic variation in diet. Similarly, a range of studies have found ontogenetic shifts in venom composition or toxicity which seem to correspond to diet shifts in the species in question [13,14,15,16], but it is typically difficult to exclude coincidental explanations in these cases. The possibility of coincidental ontogenetic shifts in both venom and diet is emphasised by studies which find a shift in one but not the other. For instance, despite sharing similar diet (differing mostly in size), juvenile and adult monocle cobras (*Naja kaouthia*) have notable differences in venom [17], while a shift from preying on lizards and frogs to preying on rodents as Gloyd’s cantil (*Agkistrodon howardgloydi*) grows is not associated with clear venom differences [18].

Some studies have related the toxicity of a venom (measured as median lethal dose, LD_50_) to different taxa that variably represent natural prey. Perhaps the best example of this work on snake venoms involves saw-scaled vipers of the genus *Echis* [19,20]. This relatively small genus contains dietary generalists, but with some species which predominantly feed on vertebrates and others which feed predominantly on arthropods [19,21]. The venom from species that prey more on arthropods is more toxic to scorpions [19], and in fact is more toxic to scorpions (of a species which they naturally prey upon) than to captive bred locusts [20], both suggesting that the venom of saw-scaled vipers has been selected for prey capture. Similar patterns have also been documented in brown tree snakes (*Boiga irregularis*), the venom of which is far more toxic to lizards and birds than to mammals [14], reflecting the focus of the diet on diapsids [22]. Furthermore, a shift to feeding on prey that does not require subjugation, such as eggs, appears to lead to reduced toxicity [11] and a general pattern of loss or degeneration of venom systems [23]. This again suggests a predominant role of diet in driving venom evolution since there is no obvious reason that, for instance, predation risk should typically change greatly following a shift of diet (as would be necessary for inferring a primarily defensive role). However, as striking and informative as these relationships are, coevolution can lead to defensive toxin resistance to snake venom in prey species [24,25], obscuring clear positive relationships between toxicity and natural diet. Moreover, given that the ultimate function of a predatory snake venom is to incapacitate rather than kill prey [26], which may not be mechanistically identical processes, the standard use of lethality as a measure of toxicity blurs the interpretation of such data from an evolutionary ecological perspective to some degree.

In addition to the above studies on whole-venom toxicity, evidence for a relationship between snake venom and diet has been documented in the form of prey-specific toxins, especially from snakes not traditionally considered to have particularly potent venom (resulting from limited activity on mammals such as humans) [27]. For instance, the three-finger toxin (3FTx) ‘denmotoxin’ was isolated from *Boiga dendrophila* venom which was ~100 times more potent on bird than mammal neurotoxicity assays in vitro [28]. The diet of this species is fairly generalist, including many mammals, but is predominantly comprised of diapsids (birds and reptiles) [29,30,31] suggesting that future study of the activity of denmotoxin on reptiles would be interesting. Nevertheless, the frequent presence of mammals in the diet of *B. dendrophila* highlights the imperfect correlation between diet and prey-specific toxins, although other mammal-specific toxins may remain to be discovered from this species. Consistent with the latter suggestion, and another example of prey-specific toxins, two 3FTxs described from the venom of *Spilotes sulphureus* have contrasting prey-specificity, with one active against mammals but not lizards, and another active against lizards but not mammals [32]. Note that prey-specific toxins are not restricted to 3FTxs; different forms of snake venom metalloproteinases (SVMPs) isolated from *Bothrops neuwiedi* have been shown to vary greatly in their toxicity to mammals vs birds [33]. Such evidence demonstrates that individual toxins may be selected to target particular prey types and provides a good mechanistic basis for how diet-driven selection of venom could operate. However, the effects of particular toxins can be difficult to extrapolate to the ecological context of how venom is used. The venom as a whole is exposed to selection only in the context of the combination of all toxins present (which may be multifunctional and/or synergistic [34]) in varying relative abundance and in various total amounts (reflecting in part venom yield). The ecological context is important to understanding the evolution of venoms in general [2,35] and should always be present in discussions of the evolutionary forces involved.

Individual caveats and limitations aside, in combination the research discussed above presents very strong evidence for diet being the predominant driver of snake venom evolution. With that as an established basis to understanding the evolutionary ecology of snake venoms, it is notable that many gaps still remain in our knowledge. The current body of work is conspicuously focused on molecular toxinology or on ‘end-point’ toxicity measures (particularly lethality measures such as LD_50_), but the intermediate steps of how those toxins lead to the incapacitation of prey is lacking from much of the literature. For instance, snake venoms can lead to several broad ‘types’ of toxicity (which we will refer to herein as ‘functional activities’ or ‘toxicological activities’) including neurotoxicity, cytotoxicity, neurotoxicity, or nephrotoxicity (note that nephrotoxicity can result from direct action of particular toxins on kidneys or an indirect but still serious consequence of other toxin targets) [9]. Attention to the evolution of these functional activities has lagged behind other levels, but is warranted for at least four reasons: (1) these functional activities are the link between molecular biology and ‘end-point’ outcomes and so can provide insight into the level of selection on venoms, (2) there is ample opportunity for selection of functional activities given that the diversity of prey eaten by snakes is likely to lead to a range of physiologies that are variably susceptible to different activities, (3) the same class of toxins (or in some cases the same toxin) can contribute to multiple forms of toxicity, and different toxins can act synergistically to lead to the incapacitation of prey, so selection on the level of functional activity may not be easily predictable from molecular or outcome studies, and (4) it is functional activities which clinicians are faced with in the event of venomous snakebite and so understanding the factors influencing these activities adds to our understanding of clinical manifestations of envenomations.

Recently, Jackson et al. [36] suggested that the type of functional activity may also covary with diet, and proposed a variety of hypotheses as potential explanations for some of their results based on proteomic work on Australian elapid snake venoms. Aside from the general hypothesis that diet should be a predictor of functional activities of venom, two of the more specific hypotheses from that paper are briefly described here. (1) Coagulotoxicity should be associated with feeding on active prey with high metabolic rates, such as endotherms and amphibians under some conditions (i.e., the raised metabolic rate of calling male frogs), because faster circulation in prey with higher metabolic rates should hasten dissemination of toxins throughout the blood volume. (2) Predation on reptiles should be associated with venoms containing more low molecular weight peptides rather than enzymatic toxins for two main reasons. Firstly, because the former may pose greater barriers for the evolution of resistance, and endogenous enzyme inhibitors from similar species (i.e., other reptiles) may make resistance to enzymatic toxins particularly easy to evolve. Secondly, because other taxa such as endotherms have relatively high and stable body temperatures, they should facilitate efficient enzyme activities, selecting for enzymatic toxic activities in snakes preying on endotherms. Herein we refer to these ideas concerning the relationship between functional activities and diet as ‘Jackson et al.’s hypotheses’.

Current evidence to support (or oppose) these hypotheses is limited in geographic and phylogenetic scope, largely restricted to a set of studies on coagulotoxicity in clades within the single radiation of Australian elapid snakes [16,37,38]. Moreover, these studies have found variable support for Jackson et al.’s hypotheses. Specialisation in reptile prey in *Pseudonaja* was associated with reduced coagulotoxicity [16], but as three out of four reptile specialists in that dataset were juveniles it is difficult to evaluate how well this explains variation across species, despite providing strong evidence primarily from intraspecific contrasts. In the Australian tigersnake clade (*Notechis* + *Hoplocephalus* + *Paroplocephalus* + *Tropidechis*), higher coagulotoxicity was associated with feeding on prey with higher metabolic rates [37], but metabolic rate of prey was scored only on a subjective 6-point scale rather than measured quantitatively, and no detectable effect of particular prey taxa was found. Although not formally tested, the pattern of coagulotoxicity in *Pseudechis* did not clearly match patterns of dietary variation across the genus [38], but the small sample size and low variation in coagulotoxicity in general prevented strong conclusions here. Each of these papers dealt only with a single type of functional activity (coagulotoxicity) and considered relationships with the extent of the activity in relatively small groups of closely related snakes, rather than comparing different activity types. Ideally evidence would take the form of a global scale comparison across large enough clades (such as caenophidian snakes) to provide appropriate variation and sample sizes and enable informative comparative analyses of the relationship between different toxicological activities of venoms and diet [39].

A final hypothesis discussed by Jackson et al. [36] is that diet breadth and venom diversity may be positively related, such that generalist predators have more complex venoms. Although not formally tested, many (but not all) of their results were consistent with this idea in Australian elapids [36]. In a different example from cone snails, venom complexity was found to be positively associated with diet breadth [40], but interestingly no one toxin superfamily was associated with a particular dietary class (vermivore vs. molluscivore vs. generalist) [40]. Nevertheless, variation in diet classes and breadth was limited (the sample contained only a single molluscivore and a single generalist species) and sample sizes for these analyses were small (10–12 species), leaving such tests with low power. It is also important to note that these studies have focused on venom complexity as measured by number of toxins or toxin classes, but functional activities may be more important since these characterise the interaction of the toxins with the prey animal’s physiology, and situations exist where a single toxin can have multiple functional activities, and multiple toxins can be involved in the same functional activity. Hence, focusing on toxin diversity instead of ‘toxicological diversity’ may mask ecologically important patterns.

Herein, we test Jackson et al.’s hypotheses on a global scale across the phylogenetic breadth of caenophidian snakes (hereafter referred to simply as ‘snakes’) with the aim of understanding how venom functional activities evolve and whether they are related to diet as predicted. Specifically, we test the following hypotheses: (1) the type of prey in a snake’s natural diet will be associated with different functional activities of the venom, with specific comparisons matching predictions from Jackson et al.’s hypotheses; (2) feeding on endothermic prey (as a strong proxy for metabolic rate) will influence functional activities of venom, particularly favouring coagulotoxicity; (3) dietary generalism will be associated with a greater diversity of venom functional activities; and (4) the type of prey in the snake’s natural diet will be associated with the toxicological diversity of the venom, since different prey classes may vary in important ways, e.g. contain different diversities of physiological targets for toxins or having different escape abilities that require faster (more complex/synergistic) immobilization of some prey types than others.

## 2. Materials and Methods

### 2.1. Data Collection

A dated phylogeny for snakes was obtained from the TimeTree database (http://timetree.org/) [41], and after pruning to match the available dataset (see below) was used in all subsequent analyses. This publicly available database of time calibrated phylogenetic trees has already been cited by over 900 publications (Google Scholar; 29th August 2019) and provides a convenient and authoritative source of large-scale phylogenetic trees for comparative analyses.

To collect standardized data on venom functional activities, diet, and body length, we used the Clinical Toxinology Resources database (http://www.toxinology.com/; hereafter CTR) [42]. CTR is a database with a clinical focus that is hosted by the Toxinology Department of the Women’s and Children’s Hospital in Adelaide, and frequently used by clinical toxicologists. We acknowledge that the CTR has a range of caveats when used for studies such as ours, but nevertheless argue that it provides a reasonable and standardized database for the traits we are considering across the global and phylogenetically diverse scales we are concerned with here. We highlight the main limitations of this dataset here and discuss why these may limit the data available to varying degrees but are unlikely to be actively misleading, and so provide a useful first large-scale test of the hypotheses considered in this paper. Firstly, and perhaps most importantly, is the clinical focus of CTR. Our data on functional activities are therefore conditional upon both sufficient information on snakebites of humans by each species as well as the responses of humans being meaningful from an evolutionary ecological perspective. In terms of sufficient information on bites, this is likely only to reduce the data availability to a subset of venomous species which have better (and perhaps therefore more reliable) information. Hence, this should not be a problem provided we still have data on a sufficient number and diversity of species to appropriately test our hypotheses. However, humans may respond differently to bites than natural prey species and so the human focus of the data has greater potential to be problematic given the evidence for prey-specific toxins discussed above. Nevertheless, although the clinical outcomes and affinity of particular toxins varies greatly with the species that is bitten, we argue that the type of response is unlikely to suffer such severe bias as at those other levels—a species that causes coagulotoxicity in prey is also likely to cause coagulotoxicity in humans, albeit with greater or lesser severity. We acknowledge that exceptions to this may exist and if symptoms from a particular functional activity are very mild in humans compared to prey species they may be overlooked, but we believe that in general this should hold (and hence that the broad-scale general patterns we are investigating should be robust to such exceptions). Secondly, it is unclear from the CTR website how frequently the database is updated; however, this should reduce the quantity of available data if information has come to light more recently, but there is no clear reason to believe the quality of the available data should suffer as a result. Finally, the biological data (e.g., diet and body size) is compiled by clinicians rather than ecologists, perhaps introducing errors due to lack of expertise in this area. Nevertheless, this information is compiled from published literature and so we expect errors to be negligible on the scale of our study. Overall therefore, we accept that better data on ecologically-relevant toxicity may become available in the future on the scale required for appropriately-powered comparative studies, and in that sense the current study is perhaps preliminary, but as this is likely to be unforthcoming in the near future we believe the CTR is sufficient for our purposes here.

We collected data on functional activities by binary coding the presence vs. absence of four main types of toxicity: cytotoxicity, neurotoxicity, nephrotoxicity, and coagulotoxicity. Note that by coding these activities individually we are not forced to simplify complex venoms into one predominant type of toxicity, but are instead able to consider the range of reported effects from venoms under these major categories. Similarly, we coded diet as the presence vs. absence of six major classes of prey: mammals, birds, reptiles, amphibians, fishes, and invertebrates. These categories were used partly for convenience (being broad enough to be comparable across all snakes) but also because they reflect major biological differences between these groups of prey that might plausibly influence the evolution of venom. Note that we considered only adult diet to avoid the confounding effect of ontogenetic shifts in diets. We also recorded total body length and included this in all models (see below) because body size is an important correlate of many aspects of ecology and physiology that could influence venom evolution [43], and has been found to correlate with venom yield [11] which could plausibly have follow on consequences for the evolution of venom composition and activity.

Once data was obtained for all species present in our original phylogeny, we restricted our dataset to species which were ‘venomous’ (clinically venomous with data on toxicity from CTR) and had data on all our variables of interest. This resulted in a final dataset of 258 species from across the phylogenetic breadth of caenophidian snakes, ~13% of total species richness [44], which we used for all subsequent analyses.

### 2.2. Data Analysis

All analyses were conducted in R 3.6.0 [45], with basic handling of the phylogenetic tree supported by the ape 5.3 package [46].

The first two sets of analyses consisted of a series of phylogenetic logistic regression models [47] implemented using the maximum penalized likelihood method in the phylolm 2.6 package [48]. First, we fit a set of four models, one with each of the functional activities as the response variable, with body length and each diet category as explanatory variables. These enabled us test whether particular diet categories predicted particular toxicological activities. Second, we fit a similar set of four models, but with the diet categories replaced with a single (binary) explanatory variable denoting whether the snake preys upon endotherms (birds and/or mammals) or not. These test whether high metabolic rate is associated with particular forms of functional activity.

Next, we fit two phylogenetic Poisson regression models based on generalized estimating equations [49], again implemented in phylolm [48], to test predictors of the diversity of toxicological activities. The response variable for both of these models was the number of functional activities (1–4) of the venom, and body length was again included as an explanatory variable in each. One of these models included the ‘diet diversity’ as an explanatory variable, measured by summing the number of diet categories (1–6). The other model we ran instead included each diet category as a separate (binary) explanatory variable. These models enabled us to test whether diet diversity or particular diet categories (respectively) predict the diversity of functional activities of the venom.

Note that these phylogenetic regression models are essentially equivalent to standard regression-style analyses and can be interpreted in the same way, but they account for the fact that we expect more closely related species to be more similar a priori and regardless of any meaningful direct relationships between their traits [39]. In this way we account for trait relationships derived only from shared ancestry and can ask whether there is an underlying relationship excluding this confounding factor. Because of the similarity to standard regression-style models, the tables of results can be interpreted in the same way, such that the coefficients are the estimated magnitude (and direction) of the relationship between the two variables, SE is the standard error of those coefficients, and z is the coefficient divided by the SE and acts as a measure of the strength of evidence (a ‘standardised effect size’) from which the P-value is derived.

The dataset and phylogeny used in the analyses for this paper are available online via the FigShare repository at https://doi.org/10.6084/m9.figshare.9724691.v2.

## 3. Results and Discussion

### 3.1. Does Type of Prey Predict Toxicological Activities of the Venom?

Very little evidence was found to support associations of prey types with the functional activities of snake venoms (Table 1). The only exceptions are that neurotoxicity is significantly more likely in reptile feeders and less likely in amphibian feeders, although these effects are modest in size and may be coincidental (Table 1), particularly for amphibians as no clear explanation for this was postulated a priori. Nevertheless, many venom neurotoxins are small peptides (e.g., 3FTxs) and so a positive association with feeding on reptiles may support Jackson et al.’s hypothesis that snakes preying upon reptiles may have venoms rich in low molecular weight toxins.

### 3.2. Do High Metabolic Rates of Prey Predict Toxicological Activities of the Venom?

Using endothermy as a (strong) proxy of metabolic rate for prey species, we find little support for metabolic rate of prey influencing toxicological activities of venoms (Table 2). In particular, we find no evidence for Jackson et al.’s [36] hypothesis that coagulotoxic venoms should be favoured by a diet consisting of prey with high metabolic rates. We note that those authors also considered amphibians to represent prey with high metabolic rates, due to the energetic costs of calling in male frogs. However, as our results do not support amphibian prey (or any other kind) being significantly associated with coagulotoxicity (Table 1) we do not believe this difference in categorization can explain our findings. We do find a positive association between nephrotoxicity and feeding on endothermic prey, though this may be an incidental effect of different kidney structures in birds and mammals compared to ectotherms. This could occur due to the unique elongated tubules within mammal and bird kidneys which may be more easily clogged by degraded protein products etc. [50,51]. Interestingly, although not significant individually, the effect size (‘coefficient’) for mammals is slightly higher than for birds (Table 1), which may support this explanation given that within a kidney, all mammal nephrons have these elongated tubules (‘loops of Henle’) whereas only a subset of bird nephrons do [50,51]. If this explanation is correct, it implies that the relationship between endotherms in the diet and nephrotoxicity may be a spandrel [52] rather than an important factor in snake venom evolution. Overall, combining these results with the last Section 3.1, we find surprisingly little evidence for prey types as a factor in the evolution of snake venom toxicological activities.

### 3.3. Does Diet Diversity Predict the Diversity of Toxicological Activities of the Venom?

In contrast to prediction of specific functional activities as discussed above, we find strong evidence for an increase in the diversity of toxicological activities with increased diet diversity (Table 3). In other words, although the specific type of toxicity can largely vary independently in relation to diet, the toxicological complexity of the venom has likely evolved in proportion to the number of different types of prey which the venom has to incapacitate. We suggest that an adaptive explanation for this lies in a greater variety of targets of different sensitivities present in the prey of highly generalist predators, and hence a greater variety of (functional) types of toxins may be more likely to act on whichever prey is envenomed at the time. This finding is broadly consistent with the results of the Jackson et al. paper which inspired the current study [36], and also with previous work looking at intraspecific variation across three populations of the cone snail *Conus miliaris* in which diet breadth, but not individual diet items, was associated with expression and differentiation of toxin genes [53].

### 3.4. Does Type of Prey Predict the Diversity of Toxicological Activities of the Venom?

In opposition to the general pattern for specific types of functional activity, their diversity in a given venom is broadly related to the types of prey found in the natural diet of snakes (Table 4). In fact, the inclusion of all taxa except amphibians was significantly related to venom toxicological complexity, and all of those taxa except invertebrates were positively related to functional diversity of venoms (such that their inclusion in the diet increased the number of functional activities of the venom). It is possible the main explanation for this is simply that by adding any particular prey type to the diet you are increasing the diversity of the diet, hence linking these results to those shown in Table 3. However, it should be noted that this is not necessarily the case since the inclusion of, for instance, mammals in the diet does not necessitate that any other prey types are retained, excluded, or added, so does not enforce an increased diet diversity in itself. Nevertheless, if the structure of real snake diets typically consists of adding a new type of prey to the pre-existing diet (without dropping another type of prey) then addition of prey type would generally have the effect of increasing overall diet diversity. This would then explain the general support found here for most prey types increasing toxicological diversity (Table 4) via the effect demonstrated in Table 3. The interesting exception to this pattern, and in fact the strongest effect in our entire study (compare the z-values in Table 1, Table 2, Table 3 and Table 4), is the negative relationship between feeding on invertebrates and toxicological diversity (Table 4). If our hypothesis to explain the patterns in Section 3.3 is correct, in that greater functional diversity derives from a greater variety of potential targets in prey, then perhaps the relative simplicity of invertebrates may favour a reduced set of toxicological activities. In particular, given the categorization of functional activities used here, the lack of kidneys and blood coagulation systems in invertebrates would immediately reduce the maximum usable functional diversity by half compared to targeting vertebrate prey. The inclusion of invertebrates in the diet may therefore lead to a streamlining of venom functional complexity.

### 3.5. General Discussion

The evolution of variation in toxicological activities of snake venoms has received surprisingly little attention, even in the context of well-known predictors of venom evolution at other levels, such as diet. Despite multifarious ecological consequences of body size variation [43], and its influence on venom yield [11], we found no effect of body length in any of our analyses. Perhaps more surprisingly, we found little overall effect of diet in predicting functional activities of venoms despite indications in a more restricted sample of Australian elapid snakes [36]. It is possible that the limitations of our dataset (see Section 2.1) prevented us from detecting such effects, but overall our study appeared sufficiently powered to detect many biological effects. Nevertheless, our results do support a role in the diversity of toxicological activities in a given venom, specifically that dietary generalism favours toxicological diversity.

This key result that ‘ecological diversity begets toxicological diversity’ in the evolution of snake venoms suggests some interesting dynamics of venom evolution. An abundance of previous literature (see Introduction) demonstrates that specific matching of venom characteristics to prey may occur at the molecular level (taxon-specific activities of toxins) and translate into venom lethality (lower LD_50_ values on typical prey items). However, our results indicate that the functional link between those two, in other words how the combination of individual toxins leads to incapacitation of prey, is not constrained to a particular toxicological route, and a variety of possible options exist for a given prey taxa. However, the diversity of physiological targets in the prey of a highly generalist snake seems to select for a diversity of toxic actions presumably to increase the chances that any of a wide range of prey are incapacitated as quickly as possible.

Considering the two more specific hypotheses from Jackson et al. [36] which we tested here, we found support for one but not the other. Specifically, we found no evidence that coagulotoxicity is associated with feeding on active prey with high metabolic rates. We suggest that the observations that stimulated this hypothesis may be atypical given that Australian elapids have evolved an unusually coagulotoxic venom composition based on recruitment of Factor Xa and (in one clade) Factor Va [54]. This does not detract from the observations within Australian elapids, which may still apply to this group, but does demonstrate the benefit of large-scale comparative studies for understanding generalisable drivers of snake venom evolution [39]. Nevertheless, our results do support the hypothesis that predation on reptiles is associated low molecular weight peptide toxins [36], given that a common form of these toxins in snake venoms are neurotoxic 3FTxs. The other side of this hypothesis is that snakes feeding on non-reptile prey should have venoms characterised by enzymatic toxins. We note that we find no results consistent with this idea but our ability to shed light on this is more limited given that enzymatic toxins are more evenly spread across toxicological functions as categorized here. Nevertheless, a recent paper suggests that venoms dominated by 3FTxs tend to occupy strongly separate areas of venom phenotypic space from enzyme-dominated venoms [55]. This may indirectly support the actual, but not necessary [55], trade-off between these two broad classes of toxin and hence venoms of non-reptile feeders might be expected to have more enzymatic toxins based on the converse of our results.

Interestingly, the explanation for venom toxicological diversity (vis a vis dietary generalism) suggested by our results may help understand some of the wide range of complexity in clinical presentations in envenomed snakebite patients [9]. A more diverse array of clinical symptoms presents a more complicated case to manage, so factors that influence the toxicological (and hence clinical) complexity may reveal insights concerning why some snakebites have worse outcomes than others. In particular, because more generalist snakes are not only likely to have more functionally diverse venoms, but also be more widely distributed [56], the most difficult snakebites to manage are also likely to occur over wide geographic distributions, synergistically contributing to the global scale of the epidemiology of snakebite envenomation [9]. Our results also suggest that despite no clear links to particular clinical syndromes (functional activities), invertebrate feeders may lead to less complex envenomations. In other words, even allowing for the greater phylogenetic distance (and hence lower similarity) of such prey to mammals such as humans, envenomations by invertebrate feeders may typically be easier to clinically manage, whether they induce neurotoxic, cytotoxic, or other symptoms.

We reiterate our earlier comments on the limitations of using clinically-oriented databases to investigate questions in evolutionary and ecological toxinology (see Section 2.1). Although we discuss reasons why this should not lead to biased results in the current manuscript, with ‘errors’ occurring in a random direction with respect to our questions and hence adding noise but no bias, we nevertheless use this limitation to highlight future directions for data collection. The ideal dataset for comparative analyses for truly ecological toxinology would consist of three types of variables. Firstly, detailed dietary (for predatory venoms) and/or predator (for defensive venoms) information that is based on extensive field (or in the case of diet information museum) studies. Such information would consist not only of a list of prey/predator species but also their relative importance in terms of frequency, selectivity (in relation to natural abundance of the prey/predators), and volumetric, numerical, or energetic importance (for prey) or relative predation risk (for predators). This information would be required at species level, not higher taxonomic levels, since the specific effects of venoms can vary substantially amongst even closely related species (though as discussed above, largely in quantitative rather than qualitative ways). Secondly, in vivo toxicological studies of the effects of venom on each prey/predator species which systematically assess the entire breadth of toxicological effects expected from the venom (the different toxicological activities as used here plus lethality and incapacitation etc.). Such data could be supplemented with in vitro studies to provide more in depth understanding of the molecular and physiological mechanisms of such effects, but ultimately the ecological relevance depends on effects of whole venoms on whole organisms, in which the relative abundance of toxins plus their interactions are important. Finally, these studies would have to be conducted across a large number of venomous species in the clade of interest to enable well-powered comparative analyses to test ecological hypotheses. Clearly such ideal datasets are neither available nor are likely to become available in the near future, hence the need to use proxies which are expected to be unbiased and enable sufficient sample sizes to reduce the effect of the added noise in our inferences, as implemented here. Nevertheless, as a wish list for future studies this outline for data collection would be a worthy goal to achieve for even one or a few well-chosen study systems, and we encourage researchers interested in ecological and evolutionary toxinology to contribute sufficient types and quality of data to work towards this (hypothetical) ideal dataset.

Overall, our results shed light on the evolution of an understudied aspect of snake venoms—variation in the type of toxicological activities—and suggest that links with diet are present but not as straightforward as with molecular toxinological studies nor studies of ‘endpoint’ measures like LD_50_. Instead of clear links between venom attributes and particular prey types, the toxicological complexity of snake venoms is more a function of the breadth of the diet. Specific types of activities can differ for a given type of prey, but dietary generalism promotes the evolution of diverse functional activities of snake venoms. Future work investigating this pattern in more detail, including links with documented ecological influences on venom complexity such as climate [12] or ecomorphological coadaptations to particular prey types [57,58] is likely to be fruitful, and we believe our current manuscript provides a new perspective to direct such studies.

## Figures and Tables

**Table 1 toxins-11-00711-t001:** Model outputs for phylogenetic logistic regression analyses testing for effects of prey type on functional activities. Significant associations are highlighted in bold.

Response Variable	Explanatory Variable	Coefficient	SE	z	P
Cytotoxicity	intercept	−1.593	1.071	−1.487	0.137
	body length	0.408	0.447	0.912	0.362
	mammals	0.390	0.450	0.867	0.386
	birds	0.255	0.282	0.904	0.366
	reptiles	−0.303	0.357	−0.847	0.397
	amphibians	−0.585	0.342	−1.709	0.087
	fishes	0.081	0.434	0.187	0.852
	invertebrates	−0.227	0.448	−0.508	0.611
Neurotoxicity	intercept	0.030	0.906	0.033	0.973
	body length	0.213	0.272	0.781	0.435
	mammals	0.062	0.264	0.236	0.813
	birds	0.158	0.192	0.822	0.411
	**reptiles**	**0.571**	**0.253**	**2.254**	**0.024**
	**amphibians**	**−0.478**	**0.222**	**−2.156**	**0.031**
	fishes	0.363	0.307	1.185	0.236
	invertebrates	−0.225	0.253	−0.891	0.373
Nephrotoxicity	intercept	−1.555	0.651	−2.388	0.017
	body length	0.415	0.457	0.908	0.364
	mammals	0.500	0.465	1.076	0.282
	birds	0.440	0.313	1.407	0.159
	reptiles	−0.369	0.397	−0.928	0.353
	amphibians	0.070	0.338	0.206	0.837
	fishes	−0.307	0.486	−0.632	0.527
	invertebrates	−0.322	0.504	−0.640	0.522
Coagulotoxicity	intercept	−0.795	0.894	−0.890	0.373
	body length	−0.007	0.359	−0.020	0.984
	mammals	−0.281	0.312	−0.903	0.366
	birds	0.280	0.222	1.264	0.206
	reptiles	0.131	0.284	0.459	0.646
	amphibians	0.384	0.261	1.471	0.141
	fishes	−0.388	0.344	−1.127	0.260
	invertebrates	−0.035	0.315	−0.111	0.912

**Table 2 toxins-11-00711-t002:** Model outputs for phylogenetic logistic regression analyses testing for effects of endothermic prey on functional activities. Significant associations are highlighted in bold.

Response Variable	Explanatory Variable	Coefficient	SE	z	P
Cytotoxicity	intercept	−1.814	0.784	−2.314	0.021
	body length	−0.146	0.466	−0.313	0.755
	endotherms	0.890	0.526	1.691	0.091
Neurotoxicity	intercept	0.697	1.178	0.592	0.554
	body length	−0.037	0.259	−0.143	0.886
	endotherms	−0.041	0.265	−0.154	0.878
Nephrotoxicity	intercept	−2.607	0.639	−4.078	4.5 × 10^−5^
	body length	0.314	0.461	0.681	0.496
	**endotherms**	**1.532**	**0.578**	**2.650**	**0.008**
Coagulotoxicity	intercept	−0.558	0.838	−0.666	0.506
	body length	−0.283	0.354	−0.801	0.423
	endotherms	0.198	0.327	0.606	0.545

**Table 3 toxins-11-00711-t003:** Model outputs for phylogenetic Poisson regression analyses testing for effects of dietary generalism on the diversity of functional activities. Significant associations are highlighted in bold.

Explanatory Variable	Coefficient	SE	z	P
intercept	0.280	0.266	1.053	0.292
body length	0.007	0.007	1.011	0.312
**diet diversity**	**0.053**	**0.017**	**3.043**	**0.002**

**Table 4 toxins-11-00711-t004:** Model outputs for phylogenetic Poisson regression analyses testing for effects of prey type on the diversity of functional activities. Significant associations are highlighted in bold.

Explanatory Variable	Coefficient	SE	z	P
intercept	0.165	0.281	0.587	0.557
body length	0.006	0.007	0.851	0.395
**mammals**	**0.212**	**0.066**	**3.191**	**0.001**
**birds**	**0.075**	**0.036**	**2.118**	**0.034**
**reptiles**	**0.128**	**0.051**	**2.480**	**0.013**
amphibians	0.056	0.045	1.244	0.213
**fishes**	**0.146**	**0.056**	**2.587**	**0.010**
**invertebrates**	**−0.405**	**0.080**	**−5.053**	**4.4 × 10^−7^**

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
