# Peer review of "Coevolution of Snake Venom Toxic Activities and Diet: Evidence that Ecological Generalism Favours Toxicological Diversity"

_toxins, 2019, doi:10.3390/toxins11120711_

Round 1

Reviewer 1 Report

It was with great interest that I began reading the present submission, focussing as it does on a subject area that I have devoted some attention to. I was not disappointed - I would firstly like to congratulate the authors on engaging with an important and neglected area of snake venom research, and in doing so through a somewhat novel lens (i.e. attempting to do it quantitatively, rather than merely qualitatively). Overall, therefore, I have no hesitation in recommending the manuscript for publication. The paper makes an important contribution to the literature on snake venom ecology. I think its criticism of certain qualitative conjectures is entirely appropriate and contributes to the process of “conjectures and refutations” celebrated by Karl Popper. This is science functioning properly. Naturally I have caveats (some of which are detailed below) about the ways in which the selected data and methods of analysis might be brought to bear on the conjectures they are being used to criticise, but these in no way diminish the value of the study.

The points about crude venom as an integrated system that is difficult to dissect are particularly apposite – we still have very little idea how various components work together inside the bodies of different target organisms under different conditions. Plenty of work for us all to do there!

I do, however, have a few minor comments: 

The authors acknowledge the limitations of the toxinology.com database for this kind of work, so I don't want to harp on about that too much. The limitations, however, are fairly profound. So much so that I feel it might be worth mentioning the data source in the abstract. The alternative approach would have been to scour the primary literature for "guts and gonads" and ecological studies that might have provided more comprehensive data on dietary composition. Having said this, I fully acknowledge that even had that large amount of work been done, it would not have changed the fact that the data would fail to be even remotely representative of the "natural facts" of snake feeding ecology. I therefore endorse the authors' pragmatism in their source of data, but feel this should be acknowledged in the abstract.  As a general comment, I'd love to see some figures in this paper, not just tables. Visual aids would greatly increase the manuscript's appeal, especially when the subject area (feeding ecology and venom) is itself so "colourful" - annotated trees would be a nice start, but even a couple of pictures of snakes feeding wouldn't go astray. Then again, it does cost money to publish colour figures, so I leave this up to the authors' discretion.  Re: the comments about the existence of a shift in venom composition but not feeding ecology in Naja kaouthia - I personally don't doubt that a lot of variation in snake venoms is epiphenomenal/neutral (I'm not going to use Gould's term "spandrel" here, since that coinage has been thoroughly debunked by Daniel Dennett) and that may be the case for the ontogenetic shift in monocled cobra venom. On the other hand, as I have noted in several papers (including the one critically interrogated in the present manuscript), there is a lot more to shifts in feeding ecology (that are potentially impactful on venom composition) than prey taxon. One factor that may play a role in the present example is the concentration-dependent effect of many toxins - the size of a prey animal may have a fairly profound effect on the suitability of certain toxins for its subjugation. Maybe this can be compensated for simply by increased venom yields correlated with increased prey size, but maybe not. Also, juvenile monocled cobra foraging behaviour may be quite different from that of adult snakes, leading to encounters with prey animals under different circumstances. Again, this may or may not have any effect on venom composition.  Along a similar line of thought - and perhaps something worth considering in a follow-up study (which I'd love to discuss) - is the fact that certain species of snakes may handle different prey taxa, or prey of the same taxa but different sizes/conditions in quite different ways. For many snakes, venom may only be important in subduing certain prey, whereas physical means (constriction, biting itself etc) may be important for other taxa. This would certainly apply to the large species of Boiga (which are basically "go anywhere, eat anything" snakes) discussed in the introduction.  In the discussion of Pseudonaja and the introduction of the "Jackson et al. hypotheses" it is mentioned that specialisation on reptilian prey was associated with reduced coaglutoxicity but that "this was difficult to separate from ontogenetic changes". In actual fact the hypotheses in question were first formulated on the basis of ontogenetic shifts in venom composition that were strongly correlated with ontogenetic shifts in feeding ecology, and the correlation of the juvenile, reptile-specialist, venom composition with that of other reptile-specialist species of Australian elapid snake. Thus, whilst I think the overall criticising of these hypotheses is entirely appropriate, the hypotheses as formulated are not quite being tested if ontogenetic shifts are excluded (as it states in the methods section they were). These are not “confounding” factors but are/were integral to the tested hypotheses. I appreciate that including them would complexify the analysis and I think the conclusions are still sound in their absence, but feel this should be noted if the study is framed as an interrogation of those particular hypotheses. As an aside, I welcome this interrogation - the function of those conjectures was to encourage exactly these sorts of criticisms and it's lovely to be part of that process.   It's also mentioned that although Jackson et al. advanced the hypothesis that generalist predators have more complex venoms, "no obvious pattern was observed that is consistent with this idea in Australian elapids". I would disagree here - indeed it was on the basis of an obvious pattern that I made this conjecture (which seems happily supported by the results of the present study). In essence, small reptile specialists (including babies of certain species and adults of others) exhibited markedly reduced venom complexity in comparison to generalists. This was correlated with the ontogenetic shift from specialisation (with simple venom) to generalisation (with more complex venom) in Pseudonaja, except in the one species of that genus (modesta) which never becomes a generalist (in the wild) and retains a less complex venom profile. An interesting discussion I’d enjoy having with the authors at a later date concerns what it means to be a genuine "specialist", versus a species that mostly eats one sort of prey due to its foraging behaviour but will eat other things if it encounters them (e.g. in captivity).  Given above comments about taxon type/class not being entirely indicative of metabolic state (poikilotherms are very different targets at different times of the day in this regard) I will question again whether the methods deployed fully test the conjectures of Jackson et al. 2016. However, I want to reaffirm that I think they test them quite nicely and pragmatically and contribute in a very meaningful way to the discussion. The observations about invertebrates and reduced venom complexity are interesting. Is this effect observed in all cases where invertebrates are included (perhaps as very occasional items) in the diet, or rather in cases where inverts become a mainstay of the diet? I appreciate that these states are hard or impossible to delineate for most species based on available data. In some cases (again see above) it may be that venomous species that occasionally feed on inverts do not in fact use their venom in subduing them (the fascinating Echis data notwithstanding). It would be nice, given the focus given to refuting certain elements of the Jackson et al hypotheses elsewhere in the paper, if the vindication of the hypothesis concerning generalist snakes having more complex venom was noted in section 2.3. Of course it goes without saying (although I seem to be saying it anyway!) that restricting the analysis to species that are “clinically venomous with data on toxicity from CTR” introduces a considerable selection bias to the dataset that moves it further away from the stated goals of the study – i.e. the investigation of the influence of feeding ecology on venom composition. A great many species of snakes use their venom in prey subjugation but are not “clinically venomous”. I know the authors are aware of this and I also appreciate how much more complex including all these species would make the study, so I do not fault them for it, I merely note that it appears to be at odds with their stated goals. Ego (further) requires me to point out that the correct order for authorship of reference number 35 is Jackson, Jouanne, and Vidal – there are some other “Jackson et al.” conjectures worth criticising in there! 

Anything else I might add (and no doubt several things I’ve already mentioned!) would be a mere quibble, so I end by again congratulating the authors on their work. If they choose to modify the manuscript in response to some of my comments I welcome that, but they have no obligation to do so.

Timothy Jackson

Reviewer 2 Report

The article proposes to evaluate, in a broad scale, the possible correlations between different types of snake venom activities and diet habits of the animals. This is a million-dollar question targeted by several toxinologists and ecologists, which is hampered by the difficulties in experimentally testing these variables in a large scale or in obtaining reliable and comparable data from other sources.

The solution adopted by the authors was to use data available in Clinical Toxinology Resources database (http://www.toxinology.com/), which is a medical information repository aimed to help in human clinics, as a proxy for natural prey envenomation effects and as a source of morphometric data. By analyzing this data (transformed to binary values models) with certain statistical tools, the relationship of venom complexity and diet diversity was evaluated.

The problems of using third part human clinical information and zoological data collected by clinicians for inferring ecological functions are so strong and obvious that the authors dedicated a whole section in the M&M to justify the choice and to explicit and discuss the limitations, which, by the way, was a quite fair inclusion provided by the authors. They justify that, despite this limitation, the dataset provides a useful first large-scale test of the hypotheses appreciated. In other words, they claim that this is the best way possible to date to test some relevant hypothesis on the field.

In my opinion, the transposition of pathophysiologic effects in humans to the different possible preys of different snake types may be severely biased. The taxonomic and morphometric data may also be not precise enough, as they recognize, and influence the analysis in a significant way. Additionally, the taxonomic range covered by the database may limit the extension of the conclusions. So the point is whether this test, even if is the only one possible, is reliable enough to provide even a primer response to the hypotheses  addressed. It is hard to tell, but my personal opinion is that it confuses more than clarify.

Nevertheless, it may be given to the readers the discretion to accept or not the evidences and conclusions, provided they are well informed about the kind of analysis performed. That is reasonably explained in the M&M, but to be truly fair in this regard and to avoid frustrated expectations, my MAJOR recommendation is to make this point clear from the beginning, as following:

1) the abstract should fully indicate that the analysis was undertaken based on clinical data. I.e., the readers should be aware that “relationships between diet and toxicological activity classes on a global scale across caenophidian snakes” investigated in the paper were not based on functional activities measured on diet items but on transposing information from clinical data. It may be rephrased in another way, but the concept should be sufficiently clear.

2) the title sounds too much assertive for me (even if the tests were on natural prey!). Something like “Testing coevolution of snake venom toxic activities and diet: ecological generalism seems to favour toxicological diversity” would be more conservative. Again, it could be written other ways but keeping compliant.

3) The limitations in this regard pointed out in M&M could be more specifically discussed along with each Result and/or broadly in the conclusionsn, since it may explain some of the lack of association observed from the analysis.

And a final suggestion is to consider that the statistical test used is not familiar to a broad audience. So perhaps some hints on what is and how to interpret “Coefficient, SE, z and P” would make easier to the reader understand the results and formulate an opinion about the findings.

Reviewer 3 Report

I enjoyed reading this paper dealing with a correlation analysis of the toxicological activities of a given venom and the type of diet to which the venomous snake is adapted. The authors found find little support for specific prey types selecting for particular toxicological effects, except that reptile-feeders are more likely to be neurotoxic. They also found find strong support for a general effect of increased diversity of prey on the diversity of toxicological effects of snake venom, hence providing evidence Hence, we provide evidence that selection pressures on the toxicological activities of snake venom has largely been driven by prey diversity rather than specific types of prey. However, correlation analyses do not include variables such as morphological co-adaptations to a specific type of prey (i.e. as suggested for the coevolution of the 3FTx-streamlined 3FTx-rich venom and skull morphology of Micrurusc surinamensis in the direction of adaptation of adaptations for an aquatic life-style and under quickly immobilization of aquatic prey); or environmental conditions (i.e. the two distinct venom phenotypes, type A (neurotoxic) and type B (hemotoxic) exhibited by Mojave rattlesnakes (Crotalus scutulatus) populations, which are geographically segregated and exhibit no discernible difference in diet. However, strong association has been found between venom type and climate, in which the neurotoxic type was found in regions with cooler winters and higher rainfall). The authors should briefly comment that including ecological constrains in a future multivariate analysis would provide more stronger support for correlating toxicological activities of snake venom and prey.

Round 2

Reviewer 2 Report

The authors defended the analysis performed, justifying by the significance of the results obtained. More importantly, they addressed my request of providing a clear from the beginning information about the data source used and of making more explicit the caveats discussed in the M&M.

Now I think the manuscript is acceptable in its present form.